# Unconventional double-bended saturation of carrier occupation in optically excited graphene due to many-particle interactions

Torben Winzer[1], Martin Mittendorff[2], Stephan Winnerl[3], Henry Mittenzwey[1], Roland Jago[4], Manfred Helm[3,5], Ermin Malic[4] & Andreas Knorr[1]

Saturation of carrier occupation in optically excited materials is a well-established phenomenon. However, so far, the observed saturation effects have always occurred in the strong-excitation regime and have been explained by Pauli blocking of the optically filled quantum states. On the basis of microscopic theory combined with ultrafast pump-probe experiments, we reveal a new low-intensity saturation regime in graphene that is purely based on many-particle scattering and not Pauli blocking. This results in an unconventional double-bended saturation behaviour: both bendings separately follow the standard saturation model exhibiting two saturation fluences; however, the corresponding fluences differ by three orders of magnitude and have different physical origin. Our results demonstrate that this new and unexpected behaviour can be ascribed to an interplay between time-dependent many-particle scattering and phase-space filling effects.

[1] Institut für Theoretische Physik, Nichtlineare Optik und Quantenelektronik, Technische Universität Berlin, Hardenbergstr. 36, 10623 Berlin, Germany. [2] Institute for Research in Electronics & Applied Physics, University of Maryland, College Park, Maryland 20742, USA. [3] Helmholtz-Zentrum Dresden-Rossendorf, P.O. Box 510119, 01314 Dresden, Germany. [4] Department of Physics, Chalmers University of Technology, SE-412 96 Gothenburg, Sweden. [5] Institut für Angewandte Physik, Technische Universität Dresden, D-01062 Dresden, Germany. Correspondence and requests for materials should be addressed to E.M. (email: ermin.malic@chalmers.se).

Saturation of light absorption is a long-known central phenomenon in nonlinear optics. It results from the fermionic character of optically driven electrons exhibiting Pauli blocking in the excited states. As a result, with increasing light fluence, the probability to increase the carrier occupation in the excited state due to light absorption is reduced. The saturation behaviour can be exploited in optical devices for the modulation of ultrashort laser pulses. In particular, mono- or multilayer graphene can be used as passive mode-locking medium[1–6]. Here the graphene sample is placed into the laser cavity[1,3] or into the output coupler[5,6], which affects the relative amplification of the frequency with highest intensity. As a result, the modulated pulse becomes sharpened in bandwidth. The advantage of graphene as a saturable absorber is a relatively low saturation intensity and an extremely short recovery time (reflecting the ultrafast carrier

dynamics[7–9]) enabling high repetition rates[1]. Furthermore, the broadband absorption of graphene allows for the modulation at different wavelengths, where the intensity can be controlled through the number of graphene layers[1,2].

The widely used approach for modelling the saturation behaviour is the two-level saturable absorber model[10–12]. It describes the saturation of the carrier occupation after continuous wave excitation accounting for the basic interplay of excitation strength, recombination, dephasing and the Pauli principle. The latter governs the high-intensity regime and explains the observed saturation behaviour. In the low-intensity regime, the Pauli blocking is negligible and the carrier occupation is proportional to the light intensity. In a strict sense, the standard saturation model can only be applied for two-level systems and continuous wave excitations. Therefore, for solid-state absorbers, such as graphene, exhibiting many electronic degrees of freedom and for ultrashort pulses produced by mode-locked solid-state lasers, the described scenario can only be of limited value.

In this work, we present a joint theory-experiment study on the saturation behaviour of the maximal optically induced carrier occupation in graphene. We apply a sophisticated theoretical approach based on graphene Bloch equations[8,13,14], which significantly go beyond the standard saturation model. The approach includes optical pumping and Coulomb- and phonon-induced many-particle carrier scattering processes on the same microscopic footing. We provide a microscopic view on the carrier dynamics in optically excited graphene and reveal a new and surprising saturation regime in the weak excitation regime that cannot be explained by the standard Pauli-blocking-based saturable absorber model. This results in a double-bended saturation of the maximal optically induced carrier occupation. The theoretical prediction is experimentally confirmed through differential transmission measurements over a large range of excitation strengths.

## Results

**Saturation behaviour.** The saturation behaviour has been mostly studied by exploiting the two-level saturable absorber model[10–12]

$$2\rho^c = \frac{I/I_s}{1 + I/I_s}, \tag{1}$$

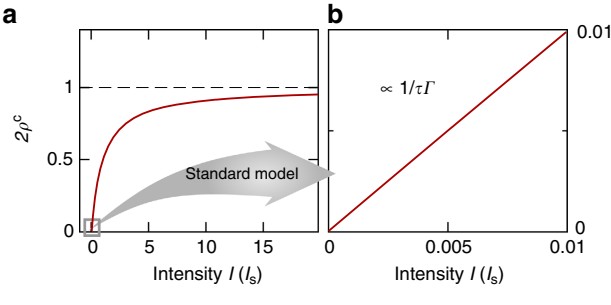

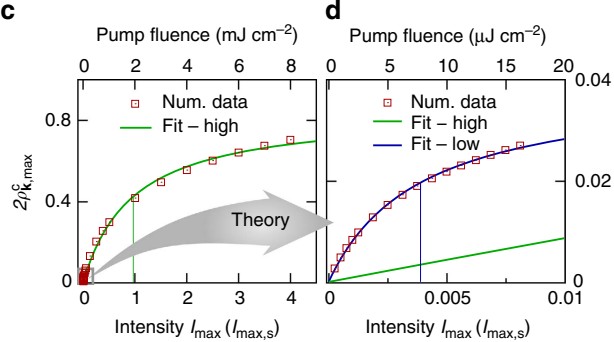

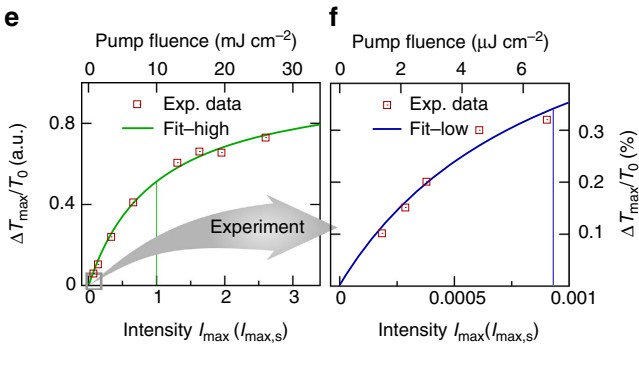

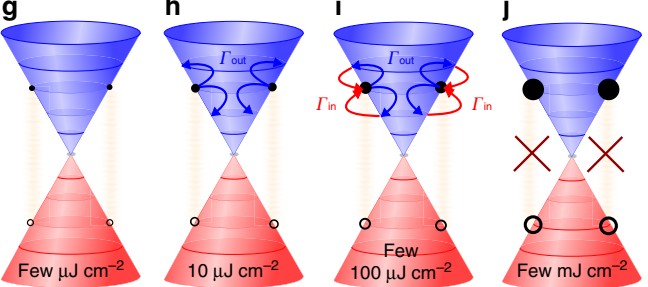

**Figure 1 | Saturation behaviour in two-level systems and graphene.**
(**a**) Saturation of the carrier occupation $\rho^c$ in a two-level system as a function of the continuous wave intensity $I$ (in units of the saturation fluence $I_s$). The maximal occupation is limited (dashed line) by Pauli blocking according to the standard saturation model in equation (1). (**b**) In the low-intensity regime of the standard saturation model the occupation is proportional to the intensity (with slope $1/\gamma\Gamma$). Theoretical prediction of the saturation of the maximal optically excited carrier occupation $\rho^c_{\mathbf{k_0},max}$ in graphene in the (**c**) high- and (**d**) low-fluence regime, respectively. The green and blue line represent the extrapolation of the high- and the low-fluence data with equation (1) yielding saturation fluences of $\mathcal{F}_s^{(h)} = 2\,\text{mJ}\,\text{cm}^{-2}$ and $\mathcal{F}_s^{(l)} = 7.8\,\mu\text{J}\,\text{cm}^{-2}$, respectively (upper abscissa). Experimental data demonstrating the saturation behaviour of the maximal differential transmission $\Delta T_{max}/T_0$ measured in the (**e**) high- and (**f**) low-excitation regime. The saturation fluences (denoted by the vertical lines) are $10\,\text{mJ}\,\text{cm}^{-2}$ and $7\,\mu\text{J}\,\text{cm}^{-2}$, respectively. (**g–j**) Schematic illustration of different saturation regimes in solid-state absorbers: (**g**) linear regime with low-scattering efficiency. (**h**) The first surprising saturation already at low intensities can be ascribed to Coulomb-induced out-scattering processes (blue arrows). (**i**) The intermediate regime, where Coulomb-induced in-scattering (red-arrows) becomes important and balances out the impact of out-scattering. (**j**) The second saturation resulting from the standard Pauli blocking of the optical excitation. The size of the filled (electrons) and open (holes) dots schematically denotes the excitation strength.

describing the saturation of the carrier occupation $\rho^c$ after continuous wave excitation with the intensity $I$. The saturation intensity $I_s$ is determined by $I_s/I = \gamma\Gamma/4\Omega^2$, with the Rabi frequency $\Omega$ ($\propto \sqrt{I}$), a constant recombination $\Gamma$ and dephasing rate $\gamma$. Obviously, equation (1) accounts for optical excitation ($\Omega$), carrier recombination ($\Gamma$) and dephasing ($\gamma$). Here Pauli blocking governs the high-intensity regime $I/I_s \gg 1$, where an increase of carrier occupation to 0.5 results in optically induced transparency, see Fig. 1a. In the low-intensity regime, the carrier occupation $\rho^c$ is proportional to the intensity $I/\gamma\Gamma$, where the slope is given by the intrinsic time scales $\gamma^{-1}$ and $\Gamma^{-1}$ of the two-level system (Fig. 1b).

The standard saturation model from equation (1) cannot be applied for solid-state absorbers with many electronic degrees of freedom nor for optical excitations with short pulses. There are fundamental differences in the electron dynamics in solids compared to the simplest saturation model in equation (1): first, for pulsed excitations, the transmission depends on time and, therefore, the maximal transmission at the energy of the optical excitation $\varepsilon_0$ and at a specific time $t_{\max}$ during an applied pulse is used to characterize the saturation behaviour. This has been applied in several recent studies investigating the transmission saturation behaviour in graphene[2,5–19]. Second, the more non-trivial electronic band structure $\varepsilon_\mathbf{k}^\lambda$ of solids (here we discuss graphene) makes the decay channels more complex, where carrier–carrier and carrier–phonon-induced in- and out-scattering for electronic states $\mathbf{k}$ (band $\lambda$) plays the crucial role rather than radiative recombination. In particular, many-particle-induced scattering channels sensitively depend on the filling of the surrounding phase space that is driven by the intensity of the excitation pulse[8,20]. This results in different regimes characterized by an interplay of time- and fluence-dependent scattering processes and phase-space filling effects.

Here we investigate the saturation behaviour of the maximal optically induced carrier occupation $\rho^c_{\mathbf{k}_0,\max}$ in graphene, which approximately determines the maximal differential transmission $\Delta T_{\max}/T_0$ measured in pump-probe experiments[8]. Here, $\mathbf{k}_0$ describes the optically excited momentum state in the conduction band. We demonstrate a new and surprising saturation regime of the carrier occupation in the excited state. The observed scenario goes beyond the standard Pauli-blocking-based saturable absorber model. We theoretically predict and experimentally confirm double-bended saturation of the maximal optically induced carrier occupation by measuring the differential transmission of a probe pulse. A first illustration of our results is shown in Fig. 1c–f. We find that the Pauli-blocking saturation is valid in the strong-excitation regime and can be fitted by the standard model from equation (1), see Fig. 1c,e for theory and experiment, respectively. For low fluences, however, where the standard model would predict a linear behaviour as shown in Fig. 1b, we find a clearly sub-linear relation between pump fluence and optically induced carrier occupation, see Fig. 1d,f for theory and experiment, respectively. Again, the curvature at low fluences follows saturation-like bending, equation (1), but the corresponding saturation fluence lies three orders of magnitude lower than in the high-fluence regime of Pauli blocking. Note that this behaviour cannot be explained by the standard Pauli-blocking saturation, since at the considered low fluences the carrier occupations $\rho^c_\mathbf{k}$ are below $10^{-2}$, suggesting a linear relation between optically induced carrier occupation and pump fluence. In contrast, our calculations reveal that the unconventional saturation-like behaviour can be ascribed to Coulomb-induced out-scattering processes that efficiently balance the pump-pulse-driven occupation in the state $\mathbf{k}_0$ (Fig. 1h). The depopulation due to Coulomb scattering at low intensity occurs at a comparable rate as the optical pumping increases the

occupation resulting in a deviation from the initial linear increase of $\rho^{c,\max}_{\mathbf{k}_0}$ with the excitation intensity. This saturation-like behaviour ceases with the increasing importance of Coulomb-induced in-scattering processes (into state $\mathbf{k}_0$) that balance out the impact of out-scattering (Fig. 1i) resulting in the regular Pauli-blocking-based saturation behaviour at high pump fluences (Fig. 1j).

**Theoretical and experimental approach.** Before we analyse the double-bended saturation and its microscopic origin in more detail, we briefly introduce our theoretical and experimental methods. The theoretical approach is based on Heisenberg equations of motion formalism combined with tight-binding wave functions[8,20–23]. To accurately model the dynamics of optically excited carriers, we account for the light–carrier interaction as well as carrier–carrier and carrier–phonon scattering on a consistent microscopic footing by solving the many-electron graphene Bloch equations. They constitute a coupled set of differential equations for the occupation probability $\rho^\lambda_\mathbf{k}$ in the state $\mathbf{k}$ in the conduction and the valence band ($\lambda = c, v$), the microscopic two-band polarization $p_\mathbf{k}$ and the phonon occupation $n^j_\mathbf{q}$ (not explicitly shown) with the momentum $\mathbf{q}$ for different optical and acoustic phonon modes $j$ (refs 7,8,13,21):

$$\dot{\rho}^c_\mathbf{k} = 2\Im\left[\Omega^*_\mathbf{k} p_\mathbf{k}\right] + \Gamma^{\rm in}_\mathbf{k}\left[1 - \rho^c_\mathbf{k}\right] - \Gamma^{\rm out}_\mathbf{k}\rho^c_\mathbf{k}, \qquad (2)$$

$$\dot{p}_\mathbf{k} = [i\Delta\omega_\mathbf{k} - \gamma_\mathbf{k}]p_\mathbf{k} - i\Omega_\mathbf{k}\left[\rho^c_\mathbf{k} - \rho^v_\mathbf{k}\right] + \mathcal{U}_\mathbf{k}, \qquad (3)$$

with the transition $\Delta\omega_\mathbf{k} = (\varepsilon^c_\mathbf{k} - \varepsilon^v_\mathbf{k})/\hbar + 2v_F|\mathbf{k}|$ and Rabi frequency $\Omega_\mathbf{k} = i\frac{e_0}{m_0}\mathbf{M}_\mathbf{k}\cdot\mathbf{A}(t)$, where $e_0$ ($m_0$) is the free electron charge (mass), $\mathbf{M}_\mathbf{k}$ the carrier–light coupling strength and $\mathbf{A}(t)$ is the vector potential of the applied laser pulse. The in- and out-scattering rates $\Gamma^{\rm in}_{\mathbf{k},\lambda}(t)$ and $\Gamma^{\rm out}_{\mathbf{k},\lambda}(t)$ contain contributions from the carrier–carrier as well as carrier–phonon interaction and depend explicitly on time and momentum, that is, implicitly on the applied pump fluence. They have been calculated microscopically within the second-order Born–Markov approximation. The total dephasing contains non-diagonal $\mathcal{U}_\mathbf{k}$ and diagonal dephasing $\gamma_\mathbf{k}$, which for symmetric conduction and valence bands reads $\gamma_\mathbf{k}(t) = \Gamma^{\rm in}_\mathbf{k}(t) + \Gamma^{\rm out}_\mathbf{k}(t)$. More details on these equations can be found in refs 7,8,13,20,21,24. The modelling is performed for an intrinsic graphene monolayer; nevertheless, the results are still valid for multilayer graphene samples, since the pump fluence variation for different layers is negligibly small compared to the studied pump fluence ranges[25].

In the following, we consider pulsed excitation and probe at 1.5 eV where the initial thermal carrier occupation can be neglected, so that the saturation of differential transmission $\Delta T_{\max}/T_0$ of the probe pulse is proportional to the maximal occupation $\rho^{c,\max}_\mathbf{k}$ in the excited state. We briefly note that compared to a previous study, where we have shown that for ultrashort excitation pulses with a duration of just 10 fs, the saturation behaviour of graphene follows the standard model (equation (1)) over eight orders of magnitude in excitation strength[9], we increase here the excitation pulse duration to $\sim 30$ fs Using these parameters, we focus in this work on the most interesting regime, where the internal time scales $1/\Gamma^{\rm in}_\mathbf{k}$ and $1/\Gamma^{\rm out}_\mathbf{k}$ are on the same order of magnitude as the pulse duration $\sigma$. In this regime, the competition between in-scattering, out-scattering and the optical pump is most important.

Degenerate pump-probe experiments were performed at room temperature on multilayer epitaxial graphene (50 layers) grown by thermal decomposition on the C-terminated face of SiC[26]. A pulsed near-infrared beam of photon energy 1.55 eV was split into a strong pump and a weaker probe beam. We measure the

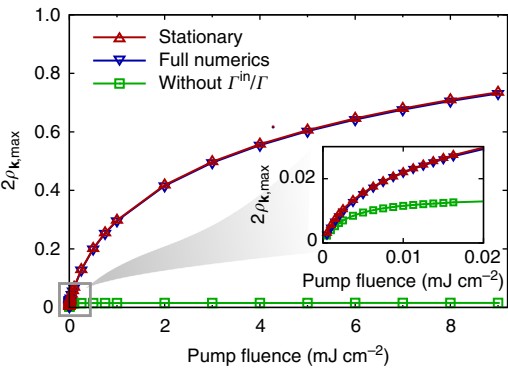

**Figure 2 | Mechanisms behind the saturation.** Saturation behaviour according to the full numerical solution of the graphene Bloch equation (blue down triangles), equations (2 and 3), and to the quasi-stationary solution (red up triangles), equation (4), show excellent agreement for all pump fluences. Without the second term $\propto \Gamma_{\mathbf{k}}^{\mathrm{in}}(t)/\gamma_{\mathbf{k}}(t)$ in equation (4), it has the same structure as the standard model and the corresponding saturation is depicted by the green squares showing a saturation in the $\mu$J cm$^{-2}$ regime and at very low carrier occupations (see the inset).

pump-induced change in transmission for varied time delays between pump and probe beam by monitoring the intensity of the probe beam with a silicon photodiode. To this end, lock-in amplification with a mechanical chopper modulating the pump beam was employed[27]. Two laser systems were used including an oscillator and an amplifier for the low and high-fluence range, respectively. The oscillator (amplifier) delivered nJ ($\mu$J) pulses of 30 fs (40 fs) duration. In all experiments, the polarization of pump and probe beam was parallel. The maximal differential transmission as a function of delay between pump and probe beam was extracted for each fluence (Fig. 1e,f). We have verified that the substrate does not contribute to the transmission via nonlinear effects in the investigated fluence range by performing experiments on the bare substrate.

**Double-bended saturation in graphene.** Now we turn to the microscopic explanation of the double-bended saturation observed in Fig. 1c–f and discuss its many-particle-induced origin as well as the resulting implications for saturation experiments. To get the basic physical picture of the elementary processes we restrict our analytical analysis in good approximation to the diagonal dephasing and derive the counterpart of equation (1) for the solid-state two-band model based on equations (2 and 3). Similar to the derivation of equation (1) for a two-level system, we apply a parametric time dependence of $\rho_{\mathbf{k}}(t)$ and $p_{\mathbf{k}}(t)$ by solving the graphene Bloch equations quasi-stationary ($\dot{\rho}_{\mathbf{k}} \approx 0$, $\dot{p}_{\mathbf{k}} \approx 0$) in rotating wave approximation, yielding

$$\frac{\Delta T(t)}{T_0} \propto 2\rho_{\mathbf{k}}^{\mathrm{c}}(t) = \frac{I(t)/I_{\mathbf{k}}(t)}{1+I(t)/I_{\mathbf{k}}(t)} + \frac{2\Gamma_{\mathbf{k}}^{\mathrm{in}}(t)/\gamma_{\mathbf{k}}(t)}{1+I(t)/I_{\mathbf{k}}(t)}, \quad (4)$$

with the time-dependent pulse intensity $I(t)$ and also the formal counterpart of the saturation intensity, denoted with $I_{\mathbf{k}}(t)$. Their ratio reads $I_{\mathbf{k}}(t)/I(t) = \gamma_{\mathbf{k}}^2(t)/4\Omega(t)^2$. Provided that the full-time dependence of the decay channels $\Gamma_{\mathbf{k}}^{\mathrm{in}}(t)$, $\Gamma_{\mathbf{k}}^{\mathrm{out}}(t)$ and $\gamma_{\mathbf{k}}(t) = \Gamma_{\mathbf{k}}^{\mathrm{in}}(t) + \Gamma_{\mathbf{k}}^{\mathrm{out}}(t)$ is considered, the quasi-stationary solution from equation (4) exhibits excellent agreement with the numerical solution of the graphene Bloch equations (2 and 3), including the saturation behaviour, see Fig. 2 illustrating the saturation according to equation (4) and to the full numerical solution (red and blue triangles).

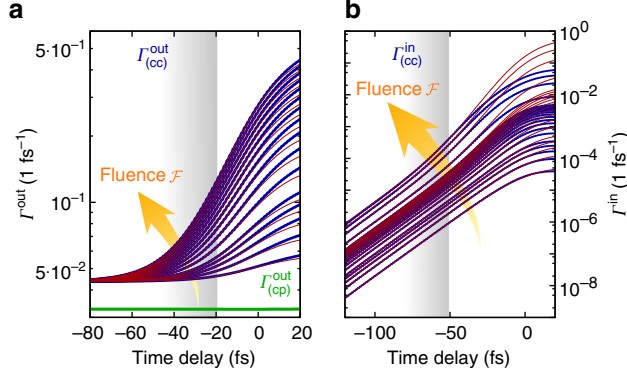

**Figure 3 | Scattering rates.** (**a**) Time and fluence dependence of the phonon- (green) and Coulomb-induced (blue) out-scattering rates. The covered fluences correspond to 0 to 9 mJ cm$^{-2}$. In good approximation, $\Gamma_{(cp)}^{\mathrm{out}}$ does not depend on time and fluence. Before the centre of the excitation pulse, the out-scattering rate $\Gamma_{(cc)}^{\mathrm{out}}$ is proportional to the time-dependent pump-fluence $\mathcal{F}(t)$ with a constant offset $\Gamma_0$ (red lines). The grey area denotes the time range, where $\Gamma_{(cc)}^{\mathrm{in}}$ can be fitted with $\propto \Gamma_0 + c\mathcal{F}(t)$ (red lines). (**b**) Time and fluence dependence of the Coulomb-induced in-scattering rates. In a time range well before the centre of the excitation pulse, $\Gamma_{(cc)}^{\mathrm{in}}$ can be fitted with $\propto c_1\mathcal{F}(t) + c_2\mathcal{F}^2(t)$ (red lines).

The saturation dynamics for the solid-state continuum, determined by equation (4), obviously differs from the two-level case in equation (1): in a solid state, in-scattering $\left(\Gamma_{\mathbf{k}_0}^{\mathrm{in}}(t)\right)$ and out-scattering processes $\left(\Gamma_{\mathbf{k}_0}^{\mathrm{out}}(t)\right)$ populating or depopulating the optically excited electronic state $\mathbf{k}_0$ are possible. For graphene, there is a linear relation between energy and momentum with $\varepsilon_0 = \hbar v_{\mathrm{F}}|\mathbf{k}_0|$. The first term in equation (4) closely resembles the standard carrier occupation saturation from equation (1), however a total rate $\gamma_{\mathbf{k}}(t)$ appears instead of a pure out-scattering contribution $\Gamma\gamma$ in the two-level case. Most importantly, an additional purely many-particle term occurs. This new contribution scaling with the ratio of the in-scattering rate $\Gamma_{\mathbf{k}}^{\mathrm{in}}(t)$ and the total scattering rate $\gamma_{\mathbf{k}}(t)$ turns out to be of crucial importance to reproduce the experimentally observed differential transmission saturation behaviour (proportional to the pump-induced carrier occupation) including the qualitative double bending as well as the quantitative saturation intensities. The neglect of this many-particle term leads to only a single saturation in the range of 3 $\mu$J cm$^{-2}$ at very low occupations (green line in Fig. 2). This means that a standard saturation model according to the first term in equation (4), that is without considering carrier in-scattering, would fail rigorously. It cannot even reproduce the standard Pauli-blocking-based saturation behaviour in the strong-excitation regime with pump fluences above 1 mJ cm$^{-2}$. Therefore, the next step is to analyse the interplay of the in- and out-scattering rates occurring in equation (4) to get a microscopic understanding of the saturation behaviour in a solid-state absorber.

The microscopically determined scattering rates $\Gamma^{\mathrm{in}}$ and $\Gamma^{\mathrm{out}}$ are plotted in Fig. 3 as a function of fluence. Focusing first on the low-excitation limit, where $\Gamma^{\mathrm{out}} \gg \Gamma^{\mathrm{in}}$, we find for the out-scattering rate a constant offset $\Gamma_0$ (owing to electron–phonon scattering) and a dominating contribution proportional to accumulated carriers with $\mathcal{F}(t) = \int_{-\infty}^{t} I(\tau)\mathrm{d}\tau$ (owing to Coulomb scattering), where $\mathcal{F}(t)$ is the fraction of the pump fluence until time $t$ (red lines in Fig. 3a). For very low pump fluence, we find $\Gamma^{\mathrm{out}} \approx \Gamma_0$, that is independent of the fluence and therefore $\rho_{\mathbf{k}_0,\max}^{\mathrm{c}}(I) \sim I$, as expected from the standard model,

equation (1). However, with increasing pump fluence, the maximal carrier occupation $\rho^c_{k_0,max}$ saturates. It turns out that due to the strong Coulomb interaction, the out-scattering rate in the Bloch equations for $\rho^c_{k_0}$ rises for increasing intensity faster than the contribution of the optical drive ($\Omega^2(t) \propto I(t)$) itself, finally balancing the optical excitation. To understand this in more detail, we consider equation (4): for the first bending, where all results can be explained without Pauli blocking, we find $\rho^c_{k_0,max}(I) \approx I(t)/I_k(t) \approx \frac{4\Omega(t)^2}{\Gamma^{out}(t)^2} = 4\Omega(t)^2/(\Gamma_0 + c\mathcal{F}(t))^2$, where $c$ is a numerical factor depending on the material properties. That is, $\rho^c_{k_0,max}(I)$ saturates due to the denominator. The interplay between the optically driven increase of the carrier occupation $\rho^c_{k_0}$ and an increasing intensity-sensitive depopulation of $\rho^c_{k_0}$ due to the efficient Coulomb-induced out-scattering (Fig. 1h) results in the first bending of $\rho^c_{k_0,max}$ in the low-fluence regime, see Fig. 1d. This means that already at low intensities, where Pauli blocking, that is the denominator in equation (4), is neglected, the fluence dependence of $\Gamma^{out}_k(I)$ starts to become important and causes the deviation from the linear dependence and thus leads to a bending. Note that this bending has nothing to do with the regular Pauli-induced saturation, since at the considered very low fluences, the carrier occupation of the excited state is below 0.01 (Fig. 1d). In contrast, our calculations reveal that the observed saturation-like behaviour is a signature of Coulomb-induced redistribution (out-scattering) at the expense of the simultaneous optical pumping in a solid-state absorber.

As shown in Fig. 2, the additional many-particle in-scattering dominated contribution (second term in equation (4)) is essential to understand the high-intensity saturation, where Pauli blocking is dominant. The dynamics of the Coulomb-induced in-scattering rate $\Gamma^{in}_{(cc)}$ is shown in Fig. 3b. We find a linear and quadratic dependence on the fluence $\mathcal{F}(t)$ with $\Gamma^{in}_{(cc)}(t) = c_1\mathcal{F}(t) + c_2\mathcal{F}^2(t)$, where $c_1$ and $c_2$ are fitting parameters, see red lines in Fig. 3b. As a result, at sufficiently high fluences, the second, in-scattering dominated term in equation (4) becomes as important as the out-scattering from $\rho^c_{k_0}$ for the saturation behaviour. At elevated intensities, the in-scattering finally balances out the loss of carrier occupation in the optically excited state $\rho^c_{k_0}$ through out-scattering processes, see Fig. 1i (detailed balance limit). In this regime, the carrier occupation increases again due to the optical pump, that is linear with the pump fluence, until a strong-excitation regime (mJ cm$^{-2}$), the Pauli-blocking regime, is reached. Here in- and out-scattering are in balance and the conventional Pauli blocking of the excited state results in a standard saturation behaviour.

## Discussion

Now, we can summarize our results from a microscopic point of view by distinguishing four distinct intensity regimes (Fig. 4a): (A) linear regime at very low pump fluences (few $\mu$J cm$^{-2}$), where the maximal optically induced carrier occupation scales linearly with the intensity $\rho^c_{k_0,max}(I) \propto I/\Gamma_0$ with nearly intensity-independent scattering rate $\Gamma_0$ (owing to electron–phonon scattering; Fig. 1g). (B) First saturation-like behaviour driven by Coulomb-induced carrier out-scattering ($\sim 10\,\mu$J cm$^{-2}$) that efficiently depopulates the optically excited state causing a deviation from the linear scaling (Fig. 1h). Here equation (4) simplifies to $\rho^c_{k_0,max}(I) \approx 4(\Omega(I)/\Gamma^{out}_k(I))^2$ and the bending results from the strong increase of the intensity-dependent out-scattering rate. (C) Intermediate regime (few 100 $\mu$J cm$^{-2}$), where the Coulomb-induced in-scattering balances out the out-scattering resulting again in an approximately linear increase of the carrier occupation due to the optical pump (Fig. 1i). (D) Second, Pauli blocking caused saturation in the high-excitation regime (few 10 mJ cm$^{-2}$; Fig. 1j). The different regimes are also recognizable

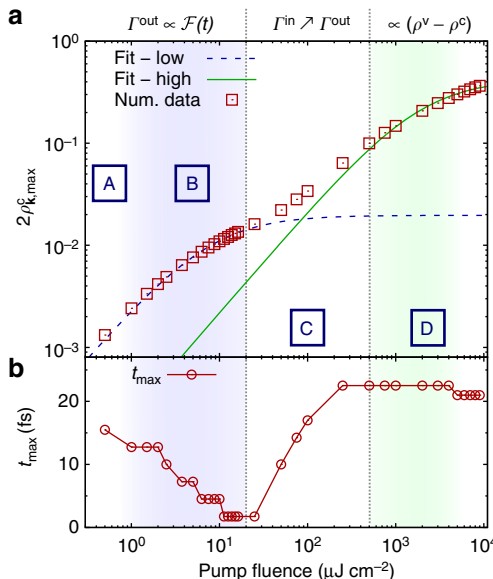

**Figure 4 | Double-bended saturation.** Four distinct regimes of saturation behaviour are as follows: (A) linear regime at very low pump fluences, (B) low-fluence out-scattering dominated regime; (C) intermediated regime, where the impact of in-scattering becomes significant and (D) the high-fluence regime, where Pauli blocking leads to a saturation. (**a**) Double-logarithmic plot of Fig. 1c showing the numerically evaluated saturation behaviour of $\rho^c_{k,max}$ (red squares) and the corresponding (blue line) low- and (green line) high-fluence fit according to equation (1). (**b**) Time delay between the centre of the excitation pulse and the time when $\rho^c_{k,max}(t)$ becomes maximal as a function of the pump fluence.

regarding the time $t_{max}$ maximizing $\rho^c_{k_0,max}(t)$ (with respect to the centre of the excitation pulse). The discrete steps of $t_{max}$ arise from our calculation beyond the rotating wave approximation and corresponds to a single-cycle duration of the exiting intensity, Fig. 4b. In the out-scattering dominated regime, $t_{max}$ shifts to earlier times due to the faster increasing out-scattering rate compared to the optical drive. Consequently, the time where out-scattering overbalances the excitation shifts towards $t_{max} \approx 0$ fs. As soon as in-scattering becomes significant the delay between pulse centre and $t_{max}$ rises again. In the Pauli-blocking regime, $t_{max}$ is constant.

In summary, based on a joint theory-experiment study, we have found a yet undiscovered double-bended saturation of the optically excited carrier occupation in graphene for optical pulse excitations having a duration in the range of the predominant scattering channels. This unconventional behaviour beyond the standard saturable absorber model is explained by the universal structure of Coulomb-induced many-particle scattering processes, clarified via in- and out-scattering and their fundamental scaling with the pump fluence. In particular, we have identified both in theory and experiment a surprising low-intensity saturation-like behaviour of the optically excited carrier occupation that is not due to Pauli blocking, but can be ascribed to the out-scattering-induced depopulation of electronic levels. Even if our accompanying analytical approach is an approximation, it covers important aspects of the dynamics. Most importantly, we find that in solids, a sub-linear relation between the optically excited carrier occupation and the pump fluence is not a sufficient criterion for an absorption saturation by means of fermionic Pauli blocking. We note that due to the universal structure of Bloch equations for solids, the observed effects are not restricted to graphene. The gained insights will be also valuable for the technological application of graphene as saturable absorber in

lasers generating ultrashort pulses[28]. Here the optically induced transmission of graphene at comparably low fluences is a specific advantage over conventional materials. Specifically, the fluences are typically in the $10\,\mu J\,cm^{-2}$ range, which corresponds to the regime of the predicted unconventional saturation behaviour.

**Data availability**. The data that support the findings of this study are available from the corresponding author upon request.

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

## Acknowledgements

We thank C. Berger and W.A. de Heer for sample fabrication and C. Habenicht for assistance in the experiments. We gratefully acknowledge support from the Deutsche Forschungsgemeinschaft (DFG) through SFB 787 (A.K.), GRK 1558 (H.M.) and SPP 1459 (S.W., T.W. and E.M.). Furthermore, this project has received funding from the European Unions Horizon 2020 research and innovation programme under grant agreement no. 696656 Graphene Flagship (E.M. and R.J.) and the EU RISE project 734690-SONAR (A.K.). Finally, we thank the Swedish Research Council VR (E.M.).

## Author contributions

T.W., H.M., R.J., E.M. and A.K. developed the theoretical model. S.W., M.M. and M.H. performed the experiments. All authors contributed to the interpretation of the results and writing of the manuscript.

## Additional information

**Competing interests:** The authors declare no competing financial interests.

**Publisher's note**: 

