## [Peer Review File · Nature Communications]

Reviewers' comments:

Reviewer #1 (Remarks to the Author):

The authors report the discovery of an unusual double-banded absorption saturation in photo-excited graphene from differential transmission data and modelling using Boltzmann equations. The two saturation regimes are attributed to the interplay between optical excitation and Coulomb-induced in- and out-scattering.

The agreement between experiment and theory is good, and the observation of a saturated absorption at two different fluences is well-supported and moderately interesting. The origin of this behavior, however, is hard to understand from the explanations given in the manuscript.

From what I understood, the authors claim that the saturation at low fluences on the order of $1\mu\text{J}/\text{cm}^2$ is due to the fact that the out-scattering rate (hence the depletion of the upper state) becomes faster than the rate at which the state is populated by the pump. Why does the absorption saturate if the upper state is efficiently depleted?

The high-fluence saturation is attributed to Pauli blocking that becomes important due to efficient in-scattering into the upper state. I understand why out-scattering of the pumped upper state is very efficient. Why should in-scattering dominate at high fluences?

I really don't understand these explanations. Maybe the authors can show the occupation probability ρ_k^λ as a function of time for different pump fluences and illustrate the phase space for in- and out-scattering for a given electronic distribution?

In summary, although the findings are moderately interesting and probably correct, the given explanation is unsatisfactory. In its present state, the manuscript is not suitable for publication.

Reviewer #2 (Remarks to the Author):

In this work the authors study the transmission through graphene of an ultrafast laser pulse via a pump-probe experiment. The main findings are that at low fluences (intensities) the transmission scales according to a law consistent with a saturation fluence that is much lower than the saturation fluence given by the Pauli-blocking limit.

The experimental results are interesting, however I find that there are several issues that in my opinion make the current version of the manuscript not suitable for publication. I list them here.

1) Not enough information is given about the experimental setup.

2) The experiments are performed on multilayer epitaxial graphene (~ 50 layers) however they are presented as performed on single layer graphene. Do the authors expect the results to be independent of the number of layers? If so, why?

3) Is the epitaxial graphene grown on SiC? Does the SiC affect the transmission?

4) The theoretical analysis is not very enlightening. The authors use a system of Bloch-equations with time dependent scattering rates to analyze the experimental data.

The number of fitting parameters is therefore infinite.

The way the scattering rates depend on time is not very well justified.

5) The main problem that I find is that from the current version of the manuscript is not clear what makes graphene to behave differently from other systems:

- Why does one need to consider time-dependent scattering rates?

- The authors state that the unconventional saturation behavior results in large part from Coulomb-induced out-scattering processes: is there a critical parameter/ratio whose value can be used to identify the conditions/systems for which the unusual saturation behavior appears?

- What is the physical origin of the "bottleneck" that limits the growth, at low fluences, of the transmission with laser intensity?

- What are the relevant physical time scales that determine the scaling of the transmission vs intensity?

- What is the pulse duration, T_p , above which the unusual saturation appears and what fixes T_p ?

In summary, I find that the manuscript contains experimental results that are potentially interesting if supported by a satisfactory physical explanation. The current version of the manuscript does not provide such explanation and so in my opinion it is not suitable for publication.

Reviewer #3 (Remarks to the Author):

Report on the manuscript NCOMMS-16-14793
by G.F. Mkrtchian

August 4, 2016

The paper by Torben Winzer, Martin Mittendorff, Stephan Winnerl, Manfred Helm, Ermin Malic, and Andreas Knorr is devoted to the problem of transmission/absorption saturation after ultrafast pulse excitation in graphene. This problem is of interest due to its fundamental applications in nanophotonics. They present theoretical as well as experimental study on this subject and reveal double-banded saturation behavior in graphene that is beyond the standard-Pauli-blocking based saturable absorber model. The theoretical model is based on the manyelectron graphene Bloch-equations

$$\begin{aligned}\frac{d\rho_{\mathbf{k}}^c}{dt} &= 2\text{Im}(\Omega_{\mathbf{k}}^* p_{\mathbf{k}}) + \Gamma_{\mathbf{k}}^{in} (1 - \rho_{\mathbf{k}}^c) - \Gamma_{\mathbf{k}}^{out} \rho_{\mathbf{k}}^c \\ \frac{dp_{\mathbf{k}}}{dt} &= [i\Delta\omega_{\mathbf{k}} - \gamma_{\mathbf{k}}] p_{\mathbf{k}} - i\Omega_{\mathbf{k}} [\rho_{\mathbf{k}}^c - \rho_{\mathbf{k}}^v] \\ \Delta\omega_{\mathbf{k}} &= (\varepsilon_{\mathbf{k}}^c - \varepsilon_{\mathbf{k}}^v) / \hbar \\ \gamma_{\mathbf{k}} &= \Gamma_{\mathbf{k}}^{in} + \Gamma_{\mathbf{k}}^{out}\end{aligned}\quad (2)$$

where they have taken into account that $\Gamma_{\mathbf{k}}^{in}$ and $\Gamma_{\mathbf{k}}^{out}$ are functions of pump wave intensities. This is the key for understanding the phenomenon reported in this paper.

The paper is suitable for publication in Nature Communications. However I have some remarks, which demand consideration before acceptance of the paper.

1) The microscopic explanation of the double-banded absorption saturation is based on the quasi-stationary solution (3) of graphene Bloch equations with the assumption $d\rho_{\mathbf{k}}^c/dt = 0, dp_{\mathbf{k}}/dt$. However, this solution can be obtained after rotating wave approximation. That is, at

$$\Omega_{\mathbf{k}} = i\tilde{\Omega}_{\mathbf{k}} e^{i\omega t} - i\tilde{\Omega}_{\mathbf{k}} e^{-i\omega t}$$

we put in Eq. (2)

$$p_{\mathbf{k}} = \tilde{p}_{\mathbf{k}} e^{i\omega t}$$

and after RWA we have

$$\frac{d\tilde{p}_{\mathbf{k}}}{dt} = [i(\Delta\omega_{\mathbf{k}} - \omega) - \gamma_{\mathbf{k}}] \tilde{p}_{\mathbf{k}} + \tilde{\Omega}_{\mathbf{k}} [\rho_{\mathbf{k}}^c - \rho_{\mathbf{k}}^v]$$

$$\frac{d\rho_{\mathbf{k}}^c}{dt} = -2\tilde{\Omega}_{\mathbf{k}}\tilde{p}_{\mathbf{k}} + \Gamma_{\mathbf{k}}^{in}(1 - \rho_{\mathbf{k}}^c) - \Gamma_{\mathbf{k}}^{out}\rho_{\mathbf{k}}^c.$$

Then at

$$d\rho_{\mathbf{k}}^c/dt = 0, d\tilde{p}_{\mathbf{k}}/dt = 0$$

for the resonance $\mathbf{k} \Rightarrow \Delta\omega_{\mathbf{k}} - \omega = 0$ we have

$$\tilde{p}_{\mathbf{k}} = \frac{\tilde{\Omega}_{\mathbf{k}}}{\gamma_{\mathbf{k}}} [\rho_{\mathbf{k}}^c - \rho_{\mathbf{k}}^v]$$

$$2\rho_{\mathbf{k}}^c = \frac{4\frac{\tilde{\Omega}_{\mathbf{k}}^2}{\gamma_{\mathbf{k}}^2}}{\frac{4\tilde{\Omega}_{\mathbf{k}}^2}{\gamma_{\mathbf{k}}^2} + 1} + 2\frac{\Gamma_{\mathbf{k}}^{in}/\gamma_{\mathbf{k}}}{\frac{4\tilde{\Omega}_{\mathbf{k}}^2}{\gamma_{\mathbf{k}}^2} + 1}$$

This is the solution (3) of the manuscript but $\tilde{\Omega}_{\mathbf{k}}$ is the amplitude of wave-graphene coupling. *Hence, the Authors should clarify this point* (if Eq. (2) is the RWA equations, then instead of $\Delta\omega_{\mathbf{k}}$ it should be $\Delta\omega_{\mathbf{k}} - \omega$ and $\Omega_{\mathbf{k}}$ should be slowly varying amplitude).

2) In this paper Authors consider absorption saturation in optically excited graphene, that is they have pump-probe experiment. For this reason absorption of probe wave is defined by the population inversion in the strong pump wave: $T \sim 2\rho_{\mathbf{k}}^c$. For the single strong pulse the absorption will be defined by the polarization:

$$\tilde{p}_{\mathbf{k}} = \frac{\tilde{\Omega}_{\mathbf{k}} \left(1 - 2\frac{\Gamma_{\mathbf{k}}^{in}}{\gamma_{\mathbf{k}}}\right)}{i(\Delta\omega_{\mathbf{k}} - \omega) - \gamma_{\mathbf{k}} - 4\frac{\tilde{\Omega}_{\mathbf{k}}^2}{\gamma_{\mathbf{k}}}}, \quad (\text{p})$$

which should be integrated over \mathbf{k} . As is seen from Eq. (p) resonance width is defined by $\gamma_{\mathbf{k}} \left(1 + 4\frac{\tilde{\Omega}_{\mathbf{k}}^2}{\gamma_{\mathbf{k}}^2}\right)$ which is intensity dependent even for constant $\gamma_{\mathbf{k}}$ and the behavior will be different from $2\rho_{\mathbf{k}}^c$. *Hence, for the concreteness I will suggest to add in the title "in optically excited graphene"*.

3) As is well known graphene as a saturable absorber is of interest for fibre lasers [1,2]. It will be better for the reader to add a text about possible applications of the revealed effect.

4) Why in fig. 1 the ranges of Pump fluences for theoretical (c) & (d) and experimental (e) & (f) data do not coincide?. For example, in Fig. 1 (c) Pump fluence- [0:9], $I_{\max}(I_{\max,s})$ - [0:5], while in 1(e) Pump fluence- [0:35], $I_{\max}(I_{\max,s})$ -[0:3.5].

[1] Bao, Q., Zhang, H., Wang, Y., Ni, Z., Yan, Y., Shen, Z. X., Loh, K. P. and Tang, D. Y. (2009), Atomic-Layer Graphene as a Saturable Absorber for Ultrafast Pulsed Lasers. *Adv. Funct. Mater.*, 19: 3077–3083. doi:10.1002/adfm.200901007.

[2] Martinez, Amos, and Zhipei Sun. "Nanotube and graphene saturable absorbers for fibre lasers." *Nature Photonics* 7.11 (2013): 842-845.

Reviewer #1

1. **Comment:** The agreement between experiment and theory is good, and the observation of a saturated absorption at two different fluences is well-supported and moderately interesting. The origin of this behavior, however, is hard to understand from the explanations given in the manuscript.

Response & Revision: We have used this critique as incentive to improve the presentation of the main message of our work. We have rewritten large parts of the manuscript including a new figure [Fig. 1(g)-(j)]. The latter illustrates the most important processes determining the saturation behaviour of optically induced carrier occupation in a solid such as graphene. The main point of rewriting is also addressed in the response to the second question.

2. **Comment:** From what I understood, the authors claim that the saturation at low fluences on the order of $1\mu\text{J}/\text{cm}^2$ is due to the fact that the out-scattering rate (hence the depletion of the upper state) becomes faster than the rate at which the state is populated by the pump. Why does the absorption saturate if the upper state is efficiently depleted?

Response: The reviewer addresses a crucial point. Due to the comments (1+2) of the referee we noticed that our discussion is not stringent: We now do not work with the misleading term absorption but we state clearly that the pump pulse induced carrier occupation saturates twice with increasing fluence due to the detailed interplay of scattering and optical pumping. These two carrier occupation saturation phenomena are observed experimentally in an absorption/transmission of the probe pulse measured via its differential transmission (with and without pump). Thus, the differential transmission saturates, not the direct absorption, since it is proportional to the pump induced carrier density. The detailed mechanism is as follows:

The origin of the low-excitation saturation-like behaviour of the optically induced carrier occupation in graphene is qualitatively different from the regular saturation stemming from Pauli blocking. Our calculations show that Coulomb-induced out-scattering processes Γ^{out}_k depleting the upper state are efficiently switched on at elevated fluence. They counteract the optical excitation that populates the state ρ^c , cf. the new Fig. 1(h) in the main manuscript. This interplay yields a transition from a linear increase of the occupation to a saturated occupation, where the optical pump process and out-scattering are in balance. To understand the interplay of out-scattering and optical pumping, it is important to emphasize that the out-scattering rate $\Gamma^{\text{out}}_k(I)$ depends on the excitation intensity I , cf. Fig. 3(a). Since the impact of out-scattering rises for increasing excitation intensity until it balances the optical excitation itself, the increase of the electron occupation at low pump fluences is followed by a bending of the occupation. We have described the corresponding processes in detail in the manuscript.

Revision: To make this point clearer, we have included the new Fig. 1(g)-(j) and we rewritten the text to a large extent, cf. the bold part of the manuscript.

3. **Comment:** The high-fluence saturation is attributed to Pauli blocking that becomes important due to efficient in-scattering into the upper state. I understand why out-scattering of the pumped upper state is very efficient. Why should in-scattering dominate at high fluences?

Response: The reviewer addresses again a crucial point clearly distinguishing the conventional saturation behavior in two-level system from the saturation in a solid. At sufficiently high fluences the second (in-scattering dominated) term in Eq. (3) becomes important for the saturation behavior. While the out-scattering rate scales linearly with the pump fluence, the in-scattering shows a quadratic dependence and is therefore of increased importance for increased fluence, cf. Fig. 3. This different behavior results from the interplay of the scattering partners and the relative importance of Pauli-blocking. A straightforward derivation of this scaling behavior can now be found in the footnote [19] in the manuscript. As a result, after the first bending of the DTS, i.e. at elevated pump fluences, the in-scattering balances the loss of carrier occupation in the optically excited state through out-scattering processes, cf. Fig. 1(i). In this intermediate regime, before the Pauli-blocking sets in, the maximal carrier occupation increases again linearly with the pump fluence, until a strong-excitation regime (mJ/cm²) is reached where the occupation approaches values close to 0.5 giving rise to Pauli-blocking and finally the standard saturation behavior for very large fluences, known from simple two-level physics

Revision: To make this point clearer, we have included the new Fig. 1(g)-(j) and we have rewritten the text to a large extent, cf. the bold part of the manuscript. The manuscript contains a detailed description of all processes.

4. **Comment:** Maybe the authors can show the occupation probability ρ_{k_0} as a function of time for different pump fluences and illustrate the phase space for in- and out-scattering for a given electronic distribution?

Response: Following the suggestion of the reviewer, we include Fig. R1 showing the resonantly pumped carrier occupation probability ρ_{k_0} in the excited state k_0 in the conduction band. One can clearly see that at low pump fluences [$1 \mu\text{J}/\text{cm}^2$] corresponding to the regime of the first saturation, the carrier occupation does not exceed values of 0.01. As a result, Pauli blocking effects can be definitely excluded as a possible mechanism for the observed saturation-like behavior. This can also be seen in Fig. 1c,d of the manuscript. In contrast, in the strong excitation regime [$1 \text{mJ}/\text{cm}^2$], the carrier occupation reaches values in the range of 0.4 Here, Pauli blocking plays a crucial role and leads to the second (standard) saturation in the optically induced carrier occupation. To understand the microscopic mechanism behind the first saturation-like behavior, time- and fluence-dependent in- and out-scattering rates are required [cf. Fig. 3], as discussed in the response to the previous comments of the reviewer.

Revision: Since time- and fluence-dependent in- and out-scattering rates present crucial insights into the saturation behavior of the optically induced carrier occupation in graphene, we have decided to keep Fig. 3 in the main manuscript and include an additional sentence describing the fluence-dependence of the carrier occupation according to Fig. R1.

Fig. R1 Time-dependent carrier occupation for different pump fluences.

Reviewer #2

- 1. Comment:** Not enough information is given about the experimental setup.

Response & Revision: We agree with the reviewer that the description of the experimental setup was too brief. In the revised version we have extended the paragraph describing the performed experiments, cf. the bold text at the beginning of page 3 in the main manuscript.

- 2. Comment:** The experiments are performed on multilayer epitaxial graphene (~50 layers), however they are presented as performed on single layer graphene. Do the authors expect the results to be independent of the number of layers? If so, why?

Response: As each layer absorbs and reflects a portion of the incoming radiation, each layer of the stack is excited by a somewhat different fluence. The pump-probe signal measured in the experiment represents an average over the different fluences. We have calculated the variation of fluence by consistently solving the Maxwell-Equations for a 50-layer graphene sample including multiple transmission and reflection according to Refs. [R1, R2]. We find that the effective pump fluence varies only about 30% from the topmost

to the lowest layer, cf. Fig. R2. This is for the case of low fluences, where no saturation is observed. For higher fluences the variation becomes smaller as the layers become more bleached. Overall, this is a small variation compared to the factor of 25 000 between the highest and the lowest fluence value applied in the differential transmission experiment. Therefore, the response of the stack is essentially similar to the response of a single layer considering the small averaging effect.

Fig. R2: The layer-dependent intensity in units of I_0 of the incoming intensity.

Revision: We have included a sentence in the main text on page 2 addressing the effect of the multilayer sample.

[R1] A. Knorr, S. Hughes, T. Stroucken, and S. Koch, *Chemical Physics* **210**, 27 (1996).

[R2] T. Stroucken, A. Knorr, P. Thomas, and S.W. Koch, *Phys. Rev. B* **53**, 2026 (1996).

3. **Comment:** Is the epitaxial graphene grown on SiC? Does the SiC affect the transmission?

Response & Revision: Indeed the investigated graphene sample is grown on SiC. This has been stated in the revised version in the new paragraph describing the performed experiments (top of page 3). SiC exhibiting a band gap of 3 eV and a Reststrahlenband between 100 and 200 meV, is fully transparent for the applied photon energy of 1.55 eV. Hence, one does not expect a pump-probe signal from the substrate. However, there is the possibility that non-linear effects give rise to signals at very high intensities. We verified experimentally that this is not the case for our fluence range by performing experiments on a bare substrate.

4. **Comment:** The theoretical analysis is not very enlightening. The authors use a system of Bloch-equations with time dependent scattering rates to analyse the experimental data. The

number of fitting parameters is therefore infinite. The way the scattering rates depend on time is not very well justified.

Response: We apply a microscopic derivation of the scattering rates via a well-established approach in solid state physics (second order Born-Markov approx.). A detailed description is given in Ref. 11,13,14. There are no fitting parameters in the description, all scattering rates are calculated without free parameters: Starting with the Heisenberg equations, we derive a coupled set of differential equations for the carrier occupation, phonon number, and microscopic polarization. The appearing time- and momentum-dependent scattering rates in these semiconductor Bloch equations have been explicitly calculated. They depend on the Coulomb matrix element and carrier occupations, cf. footnote 19 for the exact functions or Ref. 11,13,14 for a detailed description of our approach. The rates explicitly include Pauli blocking terms. They depend on time through the strongly time-dependent carrier occupations, which are calculated by numerically solving the Bloch equations including the scattering rates. As a result, our model does not rely on fitting parameters, but it is entirely microscopic. We do fix a few time-independent parameters at the beginning of the calculation (such as tight binding hopping integral, strength of the carrier-phonon interaction) through DFT-parameters (no fitting parameters!), but we do not change these parameters during the calculation, cf. Ref. 14.

Revision: We have included a sentence below Eq. 2 in the main manuscript emphasizing the microscopic footing of our approach and clarifying that the time- and momentum-dependent scattering rates have been explicitly calculated.

5. **Comment:** The main problem that I find is that from the current version of the manuscript is not clear what makes graphene to behave differently from other systems.

Response & Revision: In the revised manuscript (page 1), we have included a paragraph discussing the difference between two-level systems, where the standard saturation model from Eq. (1) is applicable, from solid-state absorbers. Graphene is only an example where we directly compare to experiments. In contrast to atomic systems, solids have a non-trivial electronic band structure resulting in more complex decay channels for the optically generated electron occupation. Here, carrier-carrier and carrier-phonon-induced in- and out-scattering for electronic states plays the crucial role rather than radiative recombination that is predominant in atomic systems. In particular, many-particle-induced scattering channels sensitively depend on the filling of the surrounding phase space that is driven by the intensity of the excitation pulse. This results in different regimes characterized by an interplay of time- and fluence-dependent scattering processes and phase space filling effects. Our predictions are not exclusive for graphene, but will also occur also in other solid-state absorbers. However, we have to state that to see these effects, our findings rely on calculation and measurements over several orders of magnitude for the incident fluence.

Also, as the referee asks, in his/her question 7+9, the proper choice for the pulse duration has to be made.

6. **Comment:** Why does one need to consider time-dependent scattering rates?

Response: Time-dependent scattering rates are determined by the microscopic theory reflecting the time-dependence of carrier occupations. This is what the microscopic theory provides. The time dependence therefore is crucial to describe the interplay between many-particle scattering processes and phase-space filling effects that take place during the investigated pulsed excitation of graphene. Taking into account the standard saturation model including constant scattering rates completely fails to explain the observed saturation behaviour.

7. **Comment:** The authors state that the unconventional saturation behavior results in large part from Coulomb-induced out-scattering processes: is there a critical parameter/ratio whose value can be used to identify the conditions/systems for which the unusual saturation behavior appears?

Response: The unconventional saturation behavior results from the time- and intensity-dependent interplay of many-body electron scattering and is therefore expected to occur also in other solid-state saturable absorbers. The important condition is pulsed excitation with pulse duration in the same order of magnitude as the material specific time scales for many-particle in- and out-scattering processes since these processes occur as competitors for the optical pump.

Revision: We denote this clearly in the manuscript.

8. **Comment:** What is the physical origin of the "bottleneck" that limits the growth, at low fluences, of the transmission with laser intensity?

Response: This is a very important point. Increasing with the induced carrier density, efficient Coulomb-induced out-scattering into optically unpumped states balances than the optical pumped state occupation. This results in a saturation-like behavior of the optically induced carrier occupation already at low pump fluences. See also the detailed response to the comment 2 of the reviewer #1 on the same question.

Revision: To make this point clearer, we have included the new Fig. 1(g)-(j) and we rewritten the text to a large extent, cf. the bold part of the manuscript.

9. **Comment:** What are the relevant physical time scales that determine the scaling of the transmission vs intensity? What is the pulse duration, T_p , above which the unusual saturation appears and what fixes T_p ?

Response: The reviewer raises another important point: The saturation behaviour is determined by the time scale of Coulomb-induced in- and out-scattering processes on one side and optical excitation on the other side. The pulse duration has to be in the same order of magnitude as the material and excitation induced specific time scales for many-particle in- and out-scattering processes. In the case of graphene, this is given for pulses in the range of 30-100fs.

Revision: This important point has been emphasized in the conclusions of our work.

Reviewer #3

1. **Comment:** The paper is suitable for publication in Nature Communications. The addressed problem is of interest due to its fundamental applications in nanophotonics.

Response: We thank the reviewer for his clear recommendation of our work for publication in Nature Communications.

2. **Comment:** The microscopic explanation of the double-banded absorption saturation is based on the quasi-stationary solution of graphene Bloch equations. However, this solution can be obtained after rotating wave approximation. Hence, the authors should clarify this point.

Response: The referee is right, for the analytical explanation, Eq. (3), we used the rotating wave approx. (RWA). However, all numerical calculations have been performed beyond the RWA, since in graphene there is no distinguished frequency that could easily justify this approximation. Nevertheless, the reviewer is completely right that our results are valid also within the RWA as long as the excitation is well above the cone intersection (then with a redefinition of the transition and Rabi frequency, as mentioned by the referee).

Revision: Below Eq. (2), we have denoted that all numerical calculations have been performed beyond the rotating wave approximation and that the analytical solution in RWA is only valid for the optical excitation well above the cone intersection.

3. **Comment:** In this paper Authors consider absorption saturation in optically excited graphene that is they have pump-probe experiment. Hence, for the concreteness I will suggest to add in the title "in optically excited graphene".

Response & Revision: We thank the reviewer for his suggestion. We have followed his advice and have changed the title to "Unconventional double-banded saturation of carrier occupation in optically excited graphene due to many-particle interactions".

4. **Comment:** As is well known graphene as a saturable absorber is of interest for fibre lasers. It will be better for the reader to add a text about possible applications of the revealed effect.

Response: In the first place, our work contributes to a better fundamental understanding of the saturation of the optically induced carrier occupation in solid-state absorbers. We show that in solids, a sub-linear relation between the maximal carrier occupation and the pump fluence is not a sufficient criterion for a saturation by the means of Pauli blocking. As the referee states, it is true that the understanding of microscopic mechanisms behind the saturation has also an impact on technological application of graphene as saturable absorber. These devices are employed in lasers generating ultrashort pulses and may be considered one of the most successful device applications of graphene in photonics. Here, the optically induced transmission of graphene at comparably low fluences is a specific advantage over other materials. Specifically, the fluences are typically in the $10 \mu\text{J}/\text{cm}^2$ range, which corresponds to the regime of the predicted unconventional saturation behavior.

Revision: We have added a paragraph in the conclusion addressing the possible impact of our findings on applications.

5. **Comment:** Why in Fig. 1 the ranges of pump fluences for theoretical (c) & (d) and experimental (e) & (f) data do not coincide?

Response: The experimental data is limited to the shown pump fluences in Fig. e,f. Our work focuses on a qualitative understanding of microscopic mechanisms behind the saturation of optically induced carrier occupation. Considering that no adjustable parameters have been used in the theory, also the obtained quantitative agreement with the experiment is very good: We find the same order of magnitude for saturation fluences both in the low excitation regime ($7.8 \mu\text{J}/\text{cm}^2$ vs. $7.0 \mu\text{J}/\text{cm}^2$) as well as in the strong excitation regime ($2 \text{ mJ}/\text{cm}^2$ vs $10 \text{ mJ}/\text{cm}^2$).

REVIEWERS' COMMENTS:

Reviewer #1 (Remarks to the Author):

The authors addressed the comments raised by all referees to my full satisfaction, resulting in a major revision of their manuscript. The microscopic origin of the double-banded absorption is much clearer now. The manuscript is suitable for publication in Nature Communications once the authors corrected a few grammatical errors and typos and other minor problems.

Minor problems:

In Figure 3 the authors write $\Gamma_0 + cF(t)$ while they write $\Gamma_0 + d_1F(t)$ in the main text. This should be consistent.

On page 5 the authors talk of a "super-linear increase of the intensity-dependent out-scattering rate". However, they use $\Gamma_{out} = \Gamma_0 + cF(t)$, which is linear in $F(t)$. This is inconsistent.

Grammar and typos (non-exhaustive):

joint theory-experiment study on the saturation of optically induced carrier occupation

interplay time-dependent many-particle scattering and phase-space filling effects

replace maximal by maximum throughout the text

the possibility of [...] is possible

however it appears a total rate

and causes [...] thus to a bending

in- and out-scattering

the 10J/cm² range

Reviewer #2 (Remarks to the Author):

I find that in the detailed response and the revised manuscript the authors address satisfactorily all the main points that were raised by the reviewers. In their response the authors mention that they have checked experimentally that for their fluence range even at relatively high intensities the substrate does not contribute to the signals via non-linear effects. I would suggest to add a short sentence to the manuscript to state that they have also done this check.

Overall I find that the revised manuscript conveys much better the results of the work and I find it suitable for publication in Nature Communications.

Reviewer #3 (Remarks to the Author):

In the revised version of the manuscript the Authors have mainly taken into account all comments and I feel that the points raised by the reviewers in the previous round of review have been satisfactorily addressed.

Thus, the paper can be accepted for publication as is.

Reviewer #1

1. **Comment:** In Figure 3 the authors write $\Gamma_{0+cF(t)}$ while they write $\Gamma_{0+d_1F(t)}$ in the main text. This should be consistent.

Response & Revision: We thank the reviewer for drawing our attention to this point. The inconsistency has been removed.

2. **Comment:** On page 5 the authors talk of a “super-linear increase of the intensity-dependent out-scattering rate”. However, they use $\Gamma_{out}=\Gamma_{0+cF(t)}$, which is linear in $F(t)$. This is inconsistent.

Response & Revision: We have removed the misleading expression “super-linear” in the revised version.

3. **Comment:** Grammar and typos

Response & Revision: We thank the reviewer for the very careful reading of our manuscript. We have removed all grammar mistakes and typos.

Reviewer #2

1. **Comment:** In their response the authors mention that they have checked experimentally that for their fluence range even at relatively high intensities the substrate does not contribute to the signals via non-linear effects. I would suggest to add a short sentence to the manuscript to state that they have also done this check.

Response & Revision: Following the suggestion of the referee, we have added a sentence clarifying the role of the substrate on page 3 (left column, above Eq. 4) of the revised manuscript.